# Shift-Robust GNNs: Overcoming the Limitations of Localized Graph Training Data

**Qi Zhu**[*]  **Natalia Ponomareva**[†]  **Jiawei Han**[*]  **Bryan Perozzi**[†]

*: University of Illinois Urbana-Champaign
†: Google Research
*{qiz3,hanj}@illinois.edu,
†{nponomareva,bperozzi}@google.com

## Abstract

There has been a recent surge of interest in designing Graph Neural Networks (GNNs) for semi-supervised learning tasks. Unfortunately this work has assumed that the nodes labeled for use in training were selected uniformly at random (i.e. are an IID sample). However in many real world scenarios gathering labels for graph nodes is both expensive and inherently biased – so this assumption can not be met. GNNs can suffer poor generalization when this occurs, by overfitting to superfluous regularities present in the training data. In this work we present a method, Shift-Robust GNN (SR-GNN), designed to account for distributional differences between biased training data and a graph's true inference distribution. SR-GNN adapts GNN models to the presence of distributional shift between the nodes labeled for training and the rest of the dataset. We illustrate the effectiveness of SR-GNN in a variety of experiments with biased training datasets on common GNN benchmark datasets for semi-supervised learning, where we see that SR-GNN outperforms other GNN baselines in accuracy, addressing at least ∼40% of the negative effects introduced by biased training data. On the largest dataset we consider, `ogb-arxiv`, we observe a 2% absolute improvement over the baseline and are able to mitigate 30% of the negative effects from training data bias [1].

## 1 Introduction

The goal of graph-based semi-supervised learning (SSL) is to use relationships between data (*its graph inductive bias*), along with a small set of labeled items, to predict the labels for the rest of a dataset. Unsurprisingly, varying exactly which nodes are labeled can have a profound effect on the generalization capability of a SSL classifier. Any bias in the sampling process to select nodes for training can create distributional differences between the training set and the rest of the graph. During inference any portion of the graph can be used, so any uneven labeling for training data can cause training and test data to have different distributions. An SSL classifier may then overfit to training data irregularities, thus hurting the performance at inference time.

Recently, GNNs have emerged as a way to combine graph structure with deep neural networks. Surprisingly, most work on semi-supervised learning using GNNs for node classification [15, 11, 1] have ignored this critical problem, and even the most recently proposed GNN benchmarks [12] assume that an independent and identically distributed (IID) sample is possible for training labels.

---

[1]Code and processed data are available at `https://github.com/GentleZhu/Shift-Robust-GNNs`.

35th Conference on Neural Information Processing Systems (NeurIPS 2021), Online.

This problem of biased training labels can be quite pronounced when GNNs are applied for semi-supervised learning in practice. It commonly happens when the size of the dataset is so large that only a subset of it can afford to be labeled – the *exact* situation where graph-based SSL is supposed to have a value proposition! While the specific source of bias can vary, we have encountered it in many different settings. For example, sometimes fixed heuristics are used to select a subset of data (which shares some characteristics) for labeling. Other times, human analysts individually choose data items for labeling, using complex domain knowledge. However, even this can be rooted in shared characteristics of data. In yet another scenario, a label source may have some latency, causing a temporal mismatch between the distribution of data at time of labeling and at the time of inference. In all of these cases, the core problem is that the GNN overfits to spurious regularities as the subset of labeled data could not be created in an IID manner.

One particular area where this can apply is in the spam and abuse domain, a common area of application for GNNs [18, 10, 33]. However, the labels in these problems usually come from explicit human annotations, which are both sparse (as human labelling is expensive), and also frequently biased. Since spam and abuse problems typically have very imbalanced label distributions (e.g. in many problems there are relatively few abusers – typically less than 1:100), labeling nodes IID results in discovering very few abusive labels. In this case choosing the points to request labels for in an IID manner is simply not a feasible option if one wants to have a reasonable number of data items from the rare class.

In this paper, we seek to quantify and address the problem of localized training data in Graph Neural Networks. We frame the problem as that of transfer learning – seeking to transfer the model's performance from a small biased portion of the graph to the entire graph itself. Our proposed framework for addressing this problem, Shift-Robust GNN (SR-GNN), strives to adapt a biased sample of labeled nodes to more closely conform to the distributional characteristics present in an IID sample of the graph. It can handle two kinds of bias that occur in both deeper GNNs and more recent linearized (shallow) versions of these models.

First we consider the case of addressing distributional shift for standard GNN models such as GCNs [15], MPNNs [7], and many more [5]. These models create deep networks which iteratively convolve information over graph structure. SR-GNN addresses this variety of distributional shift via a regularization over the hidden layers of the network. Second, we consider a class of linearized models (APPNP [16], SimpleGCN [34], etc) which decouple GNNs into non-linear feature encoding and linear message passing. These models present an interesting challenge for debiasing, as the graph can introduce bias over the features *after* all learnable layers. In cases like this, SR-GNN can use an instance reweighting paradigm to ensure that the training examples are as representative as possible over the graph data.

We illustrate the effectiveness of our proposed method on both paradigms with an experimental framework that introduces bias to the train/test split in a GNN, which lets us simulate the 'localized discovery' pattern observed in real applications on fully-labeled academic datasets. With these experiments we show that that our method SR-GNN can recover at least 40% of the performance lost when training a GCN on the same biased input.

Specifically, our contributions are the following:

1. We provide the first focused discussion on the distributional shift problem in GNNs.
2. We propose generalized framework, Shift-Robust GNN (SR-GNN), which can address shift in both shallow and deep GNNs.
3. We create an experimental framework which allows for creating biased train/test sets for graph learning datasets.
4. We run extensive experiments and analyze the results, proving that our methods can mitigate distributional shift.

## 2   Related Work

### 2.1   Distributional shift and Domain adaption work

Standard learning theory (i.e. PAC, empirical risk minimization, etc.) assumes that training and inference data is drawn from the same distribution, but there are many practical cases where this

does not hold. The question of dealing with different distributions has been widely explored as a part of the transfer learning literature. In transfer learning, the *domain adaptation* problem deals with transferring knowledge from the *source* domain (used for learning) to the *target* domain (the ultimate inference distribution).

One of the first theoretical works on domain adaptation [3] developed a distance function between a model's performance on the source and target domains to describe how similar they are. To obtain a final model, training then happens on a reweighed combination of source and target data, where weights are a function of the domain's distance. Much additional theoretical (e.g. [22]) and practical work expanded this idea and explored models which are co-trained on both source and target data. These models seek to optimize utility while minimizing the distance between extracted features distributions on both domains; this in turn led to the field of Domain Invariant Representation Learning (DIR) [6]. DIR is commonly achieved via co-training on labeled source and (unlabeled) target data. A modification to the loss either uses an adversarial head or adds additional regularizations.

More recently, various regularizations using discrepancy measures have been shown to be more stable to hyperparameters and result in better performance than adversarial heads, faster loss convergence and easier training. Maximum mean discrepancy (MMD) [19, 20] is a metric that measures difference between means of distributions in some rich Hilbert kernel space. Central moment discrepancy (CMD) [38] extends this idea and matches means and higher order moments in the original space (without the projection into the kernel). CMD has been shown to produce superior results and is less susceptible to the weight with which CMD regularization is added to the loss [20].

It is important to point out that MMD and CMD regularizations are commonly used with non-linear networks on some hidden layer (e.g. on extracted features). For linear models or non-differentiable models, prior work for domain adaptation often employed importance-reweighting instead. To find the appropriate weights, the same MMD distance was often used: in kernel mean matching (KMM) to find the appropriate weights one essentially minimizes MMD distance w.r.t the instance weights [17].

## 2.2 GNNs

Given a graph $G = \{V, E, X\}$, the nodes $V$ are associated with their features $X$ ($X \in \mathbb{R}^{|V| \times F}$) and the set of edges $E$ (*i.e.* adjacency matrix $A$, $A \in \mathbb{R}^{|V| \times |V|}$) which form connections between them. Graph neural networks [15] are neural networks that operate on both node features and graph structures. The core assumption of GNNs is that the structure of data ($A$) can provide a useful *inductive bias* for many modeling problems. GNNs output node representations $Z$ which are used for unsupervised [30] or semi-supervised [15, 11] learning. We denote the labels for SSL as $\{y_i\}$.

The general architecture $\Phi$ of a GNN consists of $K$ neural network layers which encode the nodes and their neighborhood information using some learnable weights $\theta$. More specifically the output of layer $k$ of a GNN contains a row ($h_i^k$) which can be used as a representation for each node $i$, i.e. $z_i^k = h_i^k$. Successive layers mix the node representations using graph information, for example:

$$H^k = \sigma(\tilde{A}H^{k-1}\theta^k) \tag{1}$$

where $\tilde{A}$ is an appropriately normalized adjacency matrix[2], $\sigma$ is the activation function, and $H^0 = X$. For brevity's sake, we refer to the final latent representations $Z^K$ simply as $Z$ throughout the work.

Although no existing work studies the distributional shift problem in GNNs, transfer learning of GNNs [13, 39] has explored different node-level and graph-level pre-training tasks across different graphs. Alternatively, domain adaption methods have been used to optimize a domain classifier between source and target graphs [21, 35]. In addition, distribution discrepancy minimization (MMD) has been adopted to train network embedding across domains [28]. Other regularizations for the latent state of GNNs have been proposed for domains like fairness [23]. We are the first to notice the importance of distributional shift on the same graph in a realistic setting and analyze its influence on different kinds of GNNs.

---

[2]The exact use of the adjacency matrix varies with specific GNN methods. For instance, the GCN [15] performs mean-pooling using matrix multiplication: $\tilde{A} = D^{-\frac{1}{2}}(A + I)D^{-\frac{1}{2}}$ where $I$ is the identity matrix, and $D$ is the degree matrix of $(A + I)$, $D_{ii} = \sum_j (A + I)_{ij}$.

# 3 Distributional shift in GNNs

To learn an SSL classifier, a cross-entropy loss function $l$ is commonly used,

$$\mathcal{L} = \frac{1}{M} \sum_{i=1}^{M} l(y_i, z_i), \tag{2}$$

where $z_i$ is the node representation for the node $i$ learned from a graph neural network, and $M$ is the number of training examples. When the training and testing data come from the same domain (i.e. $\text{Pr}_{\text{train}}(X, Y) = \text{Pr}_{\text{test}}(X, Y)$), optimizing the cross entropy loss over the training data ensures that a classifier is well-calibrated for performing inference on the testing data.

## 3.1 Data shift as representation shift

However, a mismatch between the training and testing distributions (*i.e.* $\text{Pr}_{\text{train}}(X, Y) \neq \text{Pr}_{\text{test}}(X, Y)$) is a common challenge for machine learning [25, 29].

In this paper, we care about the distributional shift between training and test datasets present in $Z$, the output of the last activated hidden layer. Given that the foundation of standard learning theory assumes $\text{Pr}_{\text{train}}(Y|Z) = \text{Pr}_{\text{test}}(Y|Z)$, the main cause of distribution shift is the representation shift, *i.e.* $\text{Pr}_{\text{train}}(Z, Y) \neq \text{Pr}_{\text{test}}(Z, Y) \rightarrow \text{Pr}_{\text{train}}(Z) \neq \text{Pr}_{\text{test}}(Z)$. To measure such a shift, discrepancy metrics such as MMD [8] or CMD [37] can be used. CMD measures the direct distance between distributions p and q as the following [37]:

$$\text{CMD} = \frac{1}{|b-a|} \|\text{E}(p) - \text{E}(q)\|_2 + \sum_{k=2}^{\infty} \frac{1}{|b-a|^k} \|c_k(p) - c_k(q)\|_2, \tag{3}$$

where $c_k$ is $k$-th order moment and $a, b$ denotes the joint distribution support of the distributions. In practice, only a limited number of moments is usually included (e.g. $k$=5). In this work we focus on the use of CMD [37] as a distance metric to measure distributions discrepancy for efficiency.

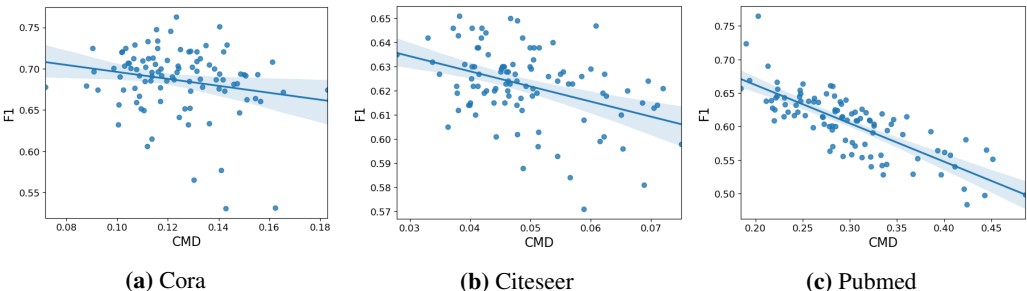

**(a)** Cora        **(b)** Citeseer        **(c)** Pubmed

**Figure 1:** Distribution shift lowers performance on GNN datasets. For each dataset, we show the performance (F1:y-axis) vs their distribution shift (CMD:x-axis) for 100 biased training set samples .

We note that GNNs (Eq (1)) are different from traditional neural networks, where the output for a layer $K$ is defined as $H^k = \sigma(H^{k-1}\theta^k)$. Instead, the multiplication of the normalized adjacency matrix ($H^k = \sigma(\tilde{A}H^{k-1}\theta^k)$) essentially changes the output distribution of the hidden representation via the graph's inductive bias. Hence, in a semi-supervised GNN, a biased training sample can lead to large representation shift due to both the graph's inductive bias in addition to 'normal' shift between non-IID sampled feature vectors.

Formally, we start the analysis of distributional shift as follows.

**Definition 3.1** (Distribution shift in GNNs). *Assume node representations $Z = \{z_1, z_2, \ldots, z_n\}$ are given as an output of the last hidden layer of a graph neural network on graph $G$ with n nodes. Given labeled data $\{(x_i, y_i)\}$ of size M, the labeled node representation $Z_l = (z_1, \ldots, z_m)$ is a subset of the nodes that are labeled, $Z_l \subset Z$. Assume $Z$ and $Z_l$ are drawn from two probability distributions p and q. The distribution shift in GNNs is then measured via a distance metric $d(Z, Z_l)$.*

Interestingly, it can be empirically shown that the effects of distribution shift due to sample bias directly lower the performance of models. To illustrate this, we plot the distribution shift distance values (x-axis) and corresponding model accuracy (y-axis) for three common GNN benchmarks using the classic GCN model [15] in Figure 1. The results demonstrate that the performance of GNNs for

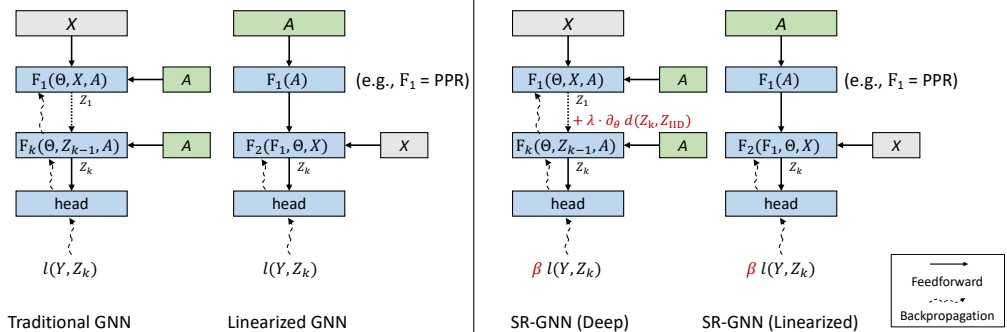

**Figure 2:** A comparison between a traditional GNN, a linearized GNN and our framework (SR-GNN).

node classification on these datasets is inversely related to the magnitude of distributional shift and motivates our investigation into distribution shift.

## 4 Shift-Robust Graph Neural Networks

In this section, we will address the distributional shift problem $(\Pr_{\text{train}}(Z) \neq \Pr_{\text{test}}(Z))$ in GNNs by proposing ways to mitigate the shift for two different GNN models (Section 4.1 and 4.2, respectively). Subsequently, we introduce SR-GNN (Fig.2) as a general framework that reduces distributional shifts for both differentiable and non-differentiable (e.g. graph inductive bias) sources simultaneously in Section 4.3.

### 4.1 Scenario 1: Traditional GNN models

We begin by considering a traditional GNN model $\Phi$, a learnable function $\mathbf{F}$ with parameters $\Theta$, over some adjacency matrix $A$:

$$\Phi = \mathbf{F}(\Theta, Z, A). \tag{4}$$

In the original GCN [15], the graph inductive bias is multiplicative at each layer and gradients are back propagated through all of the layers. In the last activated hidden layers, we denote a bounded node representation[3] as $Z \equiv Z_k = \Phi(\Theta, Z_{k-1}, A), Z_k \in [a, b]^n, Z_0 = X$.

Let us denote the training samples as $\{x_i\}_{i=1}^{M}$, the node representations are $Z_{\text{train}} = \{z_i\}_{i=1}^{M}$. For the test samples, we sample an unbiased IID sample from unlabeled data $X_{\text{IID}} = \{x_i'\}_{i=1}^{M}$ and denote the output representations as $Z_{\text{IID}} = \{z_i'\}_{i=1}^{M}$.

In order to mitigate the distributional shift between training and testing, we propose a regularizer $d : [a, b]^n \times [a, b]^n \to \mathbb{R}^+$ that is added to the cross entropy loss. Since $\Phi$ is fully differentiable, we can use a distributional shift metric as a regularization to directly minimize the discrepancy between a biased and unbiased IID sample like so:

$$\mathcal{L} = \frac{1}{M} \sum_i l(y_i, z_i) + \lambda \cdot d(Z_{\text{train}}, Z_{\text{IID}}). \tag{5}$$

Here we consider the central moment discrepancy regularizer, $d_{\text{CMD}}$:

$$d_{\text{CMD}}(Z_{\text{train}}, Z_{\text{IID}}) = \frac{1}{b-a} \|\mathbf{E}(Z_{\text{train}}) - \mathbf{E}(Z_{\text{IID}})\| + \sum_{k=2}^{\infty} \frac{1}{|b-a|^k} \|c_k(Z_{\text{train}}) - c_k(Z_{\text{IID}})\|, \tag{6}$$

where $\mathbf{E}(Z) = \frac{1}{M} \sum z_i$ and $c_k(Z) = \mathbf{E}(Z - \mathbf{E}(Z))^k$ is the k-th order moment. In practice, we use moments up to the 5th order.

---

[3]We use a bounded activation function here, *e.g.* tanh or sigmoid.

## 4.2 Scenario 2: Linearized GNN Models

Another family of recently proposed models for GNNs uses two distinct different functions – one for non-linear feature transformation, and another for a linear graph spreading stage,

$$\Phi = \mathbf{F_2}(\ \underbrace{\mathbf{F_1(A)}}_{\text{linear function}}\ , \Theta, X). \tag{7}$$

In such a linearized GNN model, the graph inductive bias is combined with node features by a linear function $\mathbf{F_1}$, which is decoupled from multi-layer neural network feature encoder $\mathbf{F_2}$. SimpleGCN [34] is an example of linearized model when $\mathbf{F_1}(A) = A^k X$. Another branch of linearized models [16, 4, 36] employs personalized pagerank to pre-compute the information diffusion in a graph (*i.e.* $\mathbf{F_1}(A) = \alpha(I - (1-\alpha)\tilde{A})^{-1}$) and apply it on encoded node features $F(\Theta, X)$.

In both models, the graph inductive bias is provided as an input feature to a linear $\mathbf{F_1}$. Unfortunately, as there are no learnable layers at this stage in these models, one can not simply apply the distributional regularizer proposed in the previous section. In this case, we can view training and testing samples as row-wise samples $h_i$ from $\mathbf{F_1}(A)$. The problem of distribution shift $\Pr_{\text{train}}(Z) \neq \Pr_{\text{test}}(Z)$ can then be transformed into matching the training and testing graph inductive bias feature space $h_i \in \mathbb{R}^n$ (where $n$ is the number of the nodes in the graph). Then to generalize from training data to testing, we can adopt an instance reweighting scheme to correct the bias, such that biased training sample $\{h_i\}_{i=1}^M$ will be similar to an IID sample $\{h_i'\}_{i=1}^M$. The resulting cross entropy loss is then

$$\mathcal{L} = \frac{1}{M} \beta_i l(y_i, \Phi(h_i)), \tag{8}$$

where $\beta_i$ be the weight for each training instance, and $l$ is the cross-entropy loss. We can then compute the optimal $\beta$ via kernel mean matching (KMM) [9] by solving a quadratic problem,

$$\min_{\beta_i} \|\frac{1}{M} \sum_{i=1}^M \beta_i \psi(h_i) - \frac{1}{M'} \sum_{i=1}^{M'} \psi(h_i')\|^2, \text{ s.t. } B_l \leq \beta < B_u \tag{9}$$

It tries to match the mean elements in a kernel $k(\cdot, \cdot)$ on the domain $\mathbb{R}^n \times \mathbb{R}^n$. Specifically, $\psi : \mathbb{R}^n \to \mathcal{H}$ denotes the feature map to the reproducing kernel Hilbert space(RKHS) introduced by kernel $k$. In our experiment, we use a mixture of gaussian kernel $k(x, y) = \sum_{\alpha_i} \exp(\alpha_i \|x - y\|_2), \alpha_i = 1, 0.1, 0.01$. The lower $B_l$ and upper bound $B_u$ constraints are there to make sure that most of the instances get some reasonable weight, as opposed to only a few instances getting non zero weight. In practice, we have multiple classes in the label space. To prevent label imbalance introduced by $\beta$, we further require that the sum of $\beta$ for a specific class $c$ remains the same before and after the correction, $\sum_i^M \beta_i \cdot \mathbb{I}(l_i = c) = \sum_i^M \mathbb{I}(l_i = c), \forall c$.

## 4.3 Shift-Robust GNN Framework

Now we propose Shift-Robust GNN (SR-GNN) - our general training objective for addressing distributional shift in GNNs:

$$\mathcal{L}_{\text{SR-GNN}} = \frac{1}{M} \beta_i l(y_i, \Phi(x_i, A)) + \lambda \cdot d(Z_{\text{train}}, Z_{\text{IID}}). \tag{10}$$

The framework consists of both a regularization for addressing distributional shift in learnable layers (Section 4.1) and an instance reweighting component which is capable of handling situations where a graph inductive bias is added after feature encoding (Section 4.2).

We will now discuss a concrete instance of our framework, by applying it to the APPNP [16] model. The APPNP model is defined as:

$$\Phi_{\text{APPNP}} = \underbrace{\left((1-\alpha)^k \tilde{A}^k + \alpha \sum_{i=0}^{k-1} (1-\alpha)^i \tilde{A}^i\right)}_{\text{approximated personalized page rank}} \underbrace{\mathbf{F}(\Theta, X)}_{\text{feature encoder}}. \tag{11}$$

It first applies a feature encoder $\mathbf{F}$ on node features $X$ and approximated personalized pagerank matrix linearly. Thereby, we have $h_i = \pi_i^{\text{ppr}}$, where $\pi_i^{\text{ppr}}$ is the personalized pagerank vector. For this, we mitigate distributional shifts from graph inductive bias via instance weighting. Moreover, let $Z = \mathbf{F}(\Theta, X)$ and we can further reduce the distributional shifts from non-linear networks by the

proposed discrepancy regularizer $d$. In our experiments, we show the application of SR-GNN on two other representative GNN models: GCN [15] and DGI [32].

# 5 Experiments

In this section we first describe how we create training set with a controllable amount of bias, then discuss our experiment design, demonstrate the efficacy our proposed framework for handling bias as well as its advantages over domain adaptation baselines, and finally, present a study on sensitivity to the hyperparameters.

## 5.1 Biased Training Set Creation

In order to study distribution shift in GNNs, we require a repeatable process which can generate graph-biased training sets. The core aspect of creating a biased sample for graph learning tasks requires an efficient method for finding 'nearby' nodes in the graph for a particular seed node. In this work, we use the Personalized PageRank (PPR) vectors to find such nearby nodes, $\Pi^{\text{ppr}} = (I - (1-\alpha)\tilde{A})^{-1}$. PPR vectors are well suited for this case for a number of reasons. First, several previous studies [36, 16] have shown strong correlations between the information diffusion in GNNs and PPR vectors. Second, a PPR vector can be computed for an individual node in time sublinear to the size of the graph [2] – so biased training samples can be easily generated, even for large datasets. This local algorithm provides a sparse approximation $\Pi^{\text{ppr}}(\epsilon)$ with guaranteed truncation error $\epsilon$, such that we can efficiently compute the top-$\gamma$ entries of a ppr vector with controllable residuals. Therefore, using PPR we can generate stable localized training data that can effectively challenge GNN models.

We obtain a biased sample from our scalable personalized pagerank sampler (PPR-S) as follows. For a certain label ratio $\tau$, we compute the number of training nodes needed per label in advance. Then we repeatedly randomly select nodes that have enough neighbors in their sparse personalized pagerank vector $\pi_i^{\text{ppr}}(\epsilon)$. We add both the seed nodes and their neighbors with the same label into the training data until we have enough number of nodes for each label.

## 5.2 Experimental settings

**Datasets.** In our experiments, we perform semi-supervised node classification tasks on five popular benchmark datasets: `Cora`, `Citeseer`, `Pubmed` [27], `ogb-arxiv` [26] and `Reddit` [11]. We use the same validation and test splits as in the original GCN paper [15] and OGB benchmark. We use the remaining nodes for training. For the unbiased baseline's performance numbers, we use a random sample from this training data. Similarly, for a biased training sample, we apply our biased sampler PPR-S on the training nodes to obtain a biased training sample and report its performance. The dataset statistics can be found in Appendix A.1.

**Baselines.** Following the two scenarios outlined in Section 4, we consider the following methods to investigate their performance under distributional shifts: (1) Traditional GNN Models: GCN [15], GAT [31], (2) Linearized GNNs: SGC [34] and APPNP [16]. We also include methods based on unsupervised node representation learning (DeepWalk [24] and DGI [32]) as a third category (3). For these methods, we use a linear classifier learned on top of pretrained node embeddings.

**Scalable biased sampler.** Our scalable biased sampler uses Personalized PageRank to efficiently create biased training samples in large graphs. Details are omitted for brevity here, but a full description can be found in Appendix A.2.

**Our Method.** If not otherwise specified, we consider the APPNP [16] instance of Shift-Robust as our base model, and also provide two ablations of it. These ablations independently use the shift-robust techniques introduced in Section 4.1 and 4.2 to validate the effectiveness of SR-GNN.

**Hyperparameters.** The main hyper parameters in our sampler PPR-S are $\alpha = 0.1, \gamma = 100$. When the graph is large, we set $\epsilon = 0.001$ in the local algorithm for sparse PPR approximation. In SR-GNN, $\lambda = 1.0$ is the penalty parameter for the discrepancy regularizer $d$, the lower bound for the instance weight $B_l$ is 0.2. For all of the GNN methods except DGI, we set the hidden dimension as 32 for Cora, Citeseer, Pubmed and 256 for ogb-arxiv, with a dropout of 0.5. In order to learn effective

**Table 1:** Semi-supervised classification on three different citation networks using biased training samples. Our proposed framework (SR-GNN) outperforms **all** baselines on biased training input.

| Method | Cora | | | Citeseer | | | PubMed | | |
|---|---|---|---|---|---|---|---|---|---|
| | Micro-F1↑ | Macro-F1↑ | ΔF1↓ | Micro-F1↑ | Macro-F1↑ | ΔF1↓ | Micro-F1↑ | Macro-F1↑ | ΔF1↓ |
| GCN (IID) | $80.8 \pm 1.6$ | $80.1 \pm 1.3$ | 0 | $70.3 \pm 1.9$ | $66.8 \pm 1.3$ | 0 | $79.8 \pm 1.4$ | $78.8 \pm 1.4$ | 0 |
| Feat.+MLP | $49.7 \pm 2.5$ | $48.3 \pm 2.2$ | 31.1 | $55.1 \pm 1.3$ | $52.7 \pm 1.3$ | 25.2 | $51.3 \pm 2.8$ | $41.8 \pm 6.2$ | 28.5 |
| Emb.+MLP | $57.6 \pm 3.0$ | $56.2 \pm 3.0$ | 23.2 | $38.5 \pm 1.2$ | $38.6 \pm 1.1$ | 31.8 | $60.4 \pm 2.1$ | $56.6 \pm 2.0$ | 19.4 |
| DGI | $71.7 \pm 4.2$ | $69.2 \pm 3.7$ | 9.1 | $62.6 \pm 1.6$ | $60.0 \pm 1.6$ | 7.6 | $58.0 \pm 5.3$ | $52.4 \pm 8.3$ | 21.8 |
| GCN | $67.6 \pm 3.5$ | $66.4 \pm 3.0$ | 13.2 | $62.7 \pm 1.8$ | $60.4 \pm 1.6$ | 7.6 | $60.6 \pm 3.8$ | $56.0 \pm 6.0$ | 19.2 |
| GAT | $58.4 \pm 5.7$ | $58.5 \pm 5.0$ | 22.4 | $58.0 \pm 3.5$ | $55.0 \pm 2.7$ | 12.3 | $55.2 \pm 3.7$ | $46.0 \pm 6.4$ | 14.6 |
| SGC | $70.2 \pm 3.0$ | $68.0 \pm 3.8$ | 10.6 | $65.4 \pm 0.8$ | $62.5 \pm 0.8$ | 4.9 | $61.8 \pm 4.5$ | $57.4 \pm 7.2$ | 18.0 |
| APPNP | $71.3 \pm 4.1$ | $69.2 \pm 3.4$ | 9.5 | $63.4 \pm 1.8$ | $61.2 \pm 1.6$ | 6.9 | $63.4 \pm 4.2$ | $58.7 \pm 7.0$ | 16.4 |
| SR-GNN **w.o.** IR | $72.1 \pm 4.4$ | $69.8 \pm 3.7$ | 8.7 | $63.9 \pm 0.7$ | $61.8 \pm 0.6$ | 6.4 | $69.4 \pm 3.4$ | $67.6 \pm 4.0$ | 10.4 |
| SR-GNN **w.o.** Reg. | $72.0 \pm 3.2$ | $69.5 \pm 3.7$ | 8.8 | $66.1 \pm 0.9$ | $63.4 \pm 0.9$ | 4.2 | $66.4 \pm 4.0$ | $64.0 \pm 5.5$ | 13.4 |
| SR-GNN (Ours) | $\mathbf{73.5 \pm 3.3}$ | $\mathbf{71.4 \pm 3.5}$ | **7.3** | $\mathbf{67.1 \pm 0.9}$ | $\mathbf{64.0 \pm 0.9}$ | **3.2** | $\mathbf{71.3 \pm 2.2}$ | $\mathbf{70.2 \pm 2.4}$ | **8.5** |

representations, DGI [32] usually needs a higher dimensional hidden space and so, following the DGI paper we set it as 512 across all of our experiments. We use Adam [14] as an optimizer, and set the learning rate to 0.01 and $L_2$ regularization to 5e-4.

**Scalability.** In this paper, we introduce two shift-robust techniques for GNN training: discrepancy regularization and instance reweighting. Let $\mathcal{O}(\Phi)$ be the time some GNN $\Phi$ takes to compute a single node embedding, and $M$ be the number of training examples. The IID sample in Eq (5) introduces $M$ extra forward passes and $2M$ extra backward propagation in total. Overall, the extra cost is therefore linear to and does not increase the existing asymptotic complexity. The Gaussian kernel computation in Eq (9) (for instance reweighting) takes $\mathcal{O}(M^2 n)$ time before training, where $h_i \in \mathbb{R}^n$. The total complexity of SR-GNN is therefore $\mathcal{O}(M\Phi + M^2 n)$. Our experiments were run on a single machine with 8 CPU and 1 Nvidia T4 GPU.

## 5.3 Experiment results

We first show the performance comparison of SR-GNN (ours) and other baselines on three well-known citation GNN benchmarks in Table 6. We report the Micro-F1, and Macro-F1 for each method. We compare each method trained on a biased sample to a GCN trained on an unbiased sample, and report its performance drop in Micro-F1 ($\Delta$F1). We begin by noting that when the training sample is biased, every method suffers a substantial performance drop (as indicated by the column $\Delta$F1). However, SR-GNN consistently reduces the influence of distributional shift and decreases the performance drop ($\Delta$F1) relative to a GCN model trained on biased input by at least 40%. We note that on these three datasets, the largest decrease in performance occurs on PubMed, where all of the existing methods experience more than a 10% absolute drop in their performance due to the biased samples. However we note that in this challenging case, the improvements of SR-GNN against APPNP (base model) also grow when the shift is larger. Finally, we see from the ablation models that the combination of both the regularization and instance reweighting appears to work better than either bias correction on its own. Our results demonstrate that our shift-robust framework is effective at minimizing the effects of distributional shift in GNN training data.

On two large benchmarks in Table 2 we see that the performance loss from biased sample is smaller but still significant. Even in a dense network like reddit, the localized training data still affect the GNN model performance. Compared with baselines, SR-GNN can effectively mitigate the 30% of the negative effect ($\Delta$) relative to an unbiased GCN. When more training data is provided (5%) we can further minimize this performance gap.

Finally, we also study how our Shift-Robust framework can be applied to two other representative models – GCNs [15], and DGI [32]. For the GCN model, we apply the regularized loss from the Equation (4) to the final node representations. For DGI, the embeddings are first trained via unsupervised learning over the entire graph. In this case, as we are optimizing a simple logistic regression over the DGI representations we can use only the instance reweighting regularization from Equation (9). Table 3 confirms that the the Shift-Robust framework successfully improves task performance in the face of biased training labels, and that it can be easily applied to a variety of different GNN models.

**Table 2:** Semi-supervised classification on ogb-arxiv and reddit varying label ratio.

| | ogb-arxiv | | | | reddit | | | |
|---|---|---|---|---|---|---|---|---|
| label(%) | 1 % | | 5 % | | 1 % | | 5 % | |
| Method | Accuracy | $\Delta\downarrow$ | Accuracy | $\Delta\downarrow$ | Accuracy | $\Delta\downarrow$ | Accuracy | $\Delta\downarrow$ |
| GCN (IID) | 66.0± 0.6 | 0 | 69.1± 0.6 | 0 | 93.8 ± 0.3 | 0 | 94.0 ± 0.1 | 0 |
| Feat.+MLP | 45.5± 0.6 | 21.5 | 43.7± 0.3 | 25.4 | 46.6±0.6 | 47.2 | 57.2±0.2 | 36.8 |
| Emb.+MLP | 51.1± 1.3 | 14.9 | 56.9± 0.8 | 13.2 | 89.6 ± 0.8 | 4.2 | 90.9 ± 0.3 | 3.1 |
| DGI | 44.8± 3.0 | 21.2 | 49.7± 3.3 | 19.4 | 83.7±1.2 | 10.1 | 85.4±0.6 | 8.6 |
| GCN | 59.3± 1.2 | 6.7 | 65.3 ± 0.6 | 3.8 | 89.7±1.0 | 4.1 | 90.9±0.3 | 3.1 |
| GAT | 58.6± 1.0 | 7.4 | 63.4 ± 1.0 | 5.7 | 80.5±5.4 | 13.3 | 82.0±3.6 | 12.0 |
| SGC | 59.0± 0.7 | 7.0 | 64.2 ± 1.3 | 4.9 | 88.6±1.0 | 5.2 | 90.6±0.2 | 3.4 |
| APPNP | 59.8± 1.1 | 6.2 | 65.1 ± 2.6 | 4.0 | 88.4±1.0 | 5.4 | 88.9±0.8 | 5.1 |
| SR-GNN **w.o.** IR | 60.6± 0.2 | 5.4 | 65.1±1.8 | 4.0 | 90.4± 0.6 | 3.4 | 91.2± 0.2 | 2.8 |
| SR-GNN **w.o.** Reg. | 61.0± 0.3 | 5.0 | 65.8±2.0 | 3.3 | 89.4± 0.8 | 4.4 | 91.9± 0.1 | 2.1 |
| SR-GNN (Ours) | **61.6±0.6** | **4.4** | **66.5±0.6** | **2.6** | **91.5± 0.5** | **2.3** | **92.1± 0.3** | **1.9** |

**Table 3:** Comparison of baseline and our SR(Shift-Robust) version ($\Delta(\%)$ -relative loss with biased sample) .

| | Cora | | | Citeseer | | | PubMed | | |
|---|---|---|---|---|---|---|---|---|---|
| Method | Micro-F1↑ | Macro-F1↑ | $\Delta(\%)$ | Micro-F1↑ | Macro-F1↑ | $\Delta(\%)$ | Micro-F1↑ | Macro-F1↑ | $\Delta(\%)$ |
| GCN (IID) | 80.8 | 80.1 | 0% | 70.3 | 66.8 | 0% | 79.8 | 78.8 | 0% |
| GCN | 67.6 | 66.4 | -12% | 62.7 | 60.4 | -8% | 60.6 | 56.0 | -19% |
| SR-GCN | **69.6** | **68.2** | -10% | **64.7** | **62.0** | -6% | **67.0** | **65.2** | -13% |
| DGI (IID) | 80.6 | 79.3 | 0% | 70.8 | 66.7 | 0% | 77.6 | 77.0 | 0% |
| DGI | 71.7 | 69.2 | -9% | 62.6 | 60.0 | -8% | 58.0 | 52.4 | -20% |
| SR-DGI | **74.3** | **72.6** | -6% | **65.8** | **62.6** | -6% | **62.0** | **57.8** | -16% |

## 5.4 Comparison with other domain invariant learning methods

In SR-GNN, we utilize the unlabeled data $Z_{\text{IID}}$ sampled from the unifying set of training and test dataset. In domain invariant learning [6, 37], unlabeled data from target domain is also used to regularize latent space between source and target domain. DANN [6] is a method that uses an adversarial domain classifier to encourage similar feature distributions between different domains. In our case, the domains are a biased training data (source) and IID unlabeled data (target). We compare the CMD regularizer and DANN regularizer using the same GNN architectures (*i.e.* GCN [15], APPNP [16]). The hyper parameter $\lambda$ is used to weight the domain loss and the CMD regularization respectively. We tune this hyper parameter on the validation set and report models' performance under the best $\lambda$ (1 for Cora and Citeseer, 0.1 for PubMed). In Table 4, as discussed in the related work, the discrepancy measures generally perform better than adversarial domain invariant learning (and CMD has been shown to be less sensitive to the regularizer weight). Especially under semi-supervised setting, the performance of DANN is more sensitive to the domain loss. Notice that CMD (Ours) is the ablation (SR-GNN **w.o.** IR) in the main paper – we note that we have already shown its performance can be further boosted using instance reweighting.

**Table 4:** Comparison of Domain-Adversarial Neural Network (DANN) and CMD regularizer used in SR-GNN with biased training data.

| | Cora | | Citeseer | | PubMed | |
|---|---|---|---|---|---|---|
| Method | Micro-F1↑ | Macro-F1↑ | Micro-F1↑ | Macro-F1↑ | Micro-F1↑ | Macro-F1↑ |
| GCN | 68.3 | 67.2 | 62.4 | 60.2 | 59.2 | 53.8 |
| DANN | 69.8 | 68.5 | 63.8 | 61.0 | 64.8 | 61.8 |
| CMD (Ours) | **71.0** | **69.4** | **65.0** | **62.3** | **67.5** | **66.2** |
| APPNP | 71.3 | 69.2 | 63.9 | 61.6 | 64.8 | 60.4 |
| DANN | 71.6 | 69.5 | 64.3 | 61.8 | 67.8 | 65.4 |
| CMD (Ours) | **72.4** | **70.1** | **65.0** | **62.4** | **70.4** | **68.7** |

## 5.5 Parameter sensitivity of SR-GNN

In this section we study how varying some of our hyper-parameters affects the performance. We perform 20 runs for each parameter and fix the initialization for all the models per run. Studies on SR-GNN with more layers (deeper models) and different biased samples are in Appendix A.3.

**Performance with different $\alpha$ in PPR-S.** Previously, we set $\alpha = 0.1$ in the biased sampler PPR-S. In Figure 3, we vary $\alpha$ between $[0.05, 0.3]$ to evaluate the performance of APPNP and SR-GNN as the bias of the sample changes. While the absolute accuracy varies with different $\alpha$, SR-GNN consistently beats its base model by a clear margin. The different patterns across three datasets are expected, since as $\alpha$ grows, the PPR-neighbors can have different topological properties and structure.

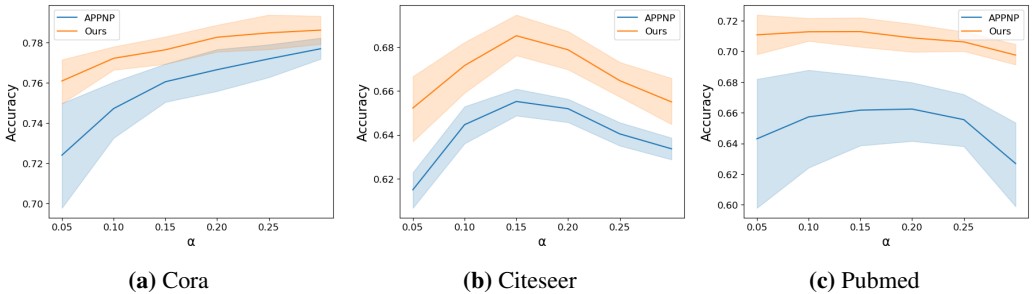

| (a) Cora | (b) Citeseer | (c) Pubmed |

**Figure 3:** Varying $\alpha$ of biased sampler on three benchmarks.

**Performance with different $B_l$, $\lambda$ in SR-GNN.** In our framework, there are three major hyper parameters: the regularization strength $\lambda$, the number of moments used in CMD $k$, and the bounds of instance weights: $B_l, B_u$. In Figure 4a, the performance of SR-GNN with different $k$ and $\lambda$ is reported. Even when only one moment is used for the CMD estimation, the method can still perform well with a reasonable penalty $\lambda$. We study different choices of $B_l$ (for a constant $B_u$) in Figure 4b. The result shows a smaller $B_l$ is better for training since larger values limit the expressive range.

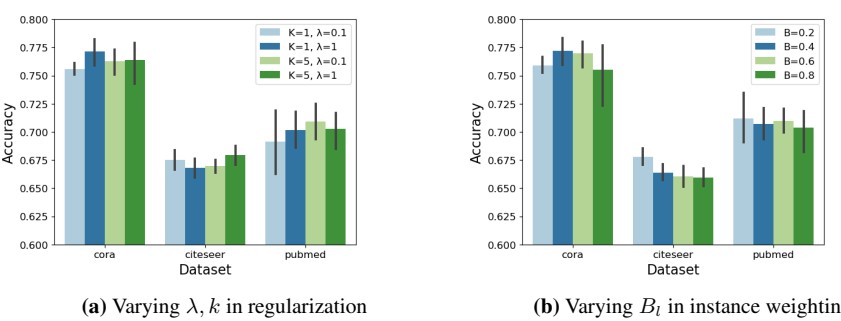

| (a) Varying $\lambda, k$ in regularization | (b) Varying $B_l$ in instance weighting |

**Figure 4:** Parameter sensitivity of SR-GNN.

## 6 Conclusion

In this paper we were the first to demonstrate that unbiased training data is very important for performance of GNNs. We argued that biased training data is extremely common in real world scenarios and can arise due to a variety of reasons including: difficulties of labelling large amount of data, the various heuristics or inconsistent techniques that are used to choose nodes for labelling, delayed label assignment, and other constraints from real world problems. We presented a general framework (SR-GNN) that is able to reduce influence of biased training data and can be applied to various types of GNNs, including both deeper GNNs and more recent linearized (shallow) versions of these models. With a number of experiments, we demonstrated both GNNs susceptibility to biased data and the success of our method in mitigating performance drops due to this bias: our method outperforms other GNN baselines on biased data and eliminates between $(30-50\%)$ of the negative effects introduced by training a GCN on biased training data.

However, there is still much to do. For instance, while we have considered the general problem of distributional shift in GNNs, there is much specific work that can (and should) be done in specific domains! Future work in this area should include regularizations to maximize performance for particular kinds of distribution shift (e.g. in spam & abuse detection) and to ensure constraints (e.g. fairness) in the presence of imbalanced training data.

## Acknowledgments and Disclosure of Funding

Research was supported in part by US DARPA KAIROS Program No. FA8750-19-2-1004, SocialSim Program No. W911NF-17-C-0099, and INCAS Program No. HR001121C0165, National Science Foundation IIS-19-56151, IIS-17-41317, and IIS 17-04532, and the Molecule Maker Lab Institute: An AI Research Institutes program supported by NSF under Award No. 2019897. Any opinions, findings, and conclusions or recommendations expressed herein are those of the authors and do not necessarily represent the views, either expressed or implied, of DARPA or the U.S. Government. We would like to thank AWS Machine Learning Research Awards program for providing computational resources for the experiments in this paper.

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
