# OpenReview forum: "Shift-Robust GNNs: Overcoming the Limitations of Localized Graph Training data"
_NeurIPS.cc/2021/Conference — NeurIPS 2021 Poster_

### Official Review · Reviewer_6hbj · 2021-07-15

**Rating:** 6
**Confidence:** 3

**Summary:**

This paper proposes the Shift-Robust GNNs (SR-GNN) which provide the robust graph neural network model under the distribution shift scenario where the data distributions of train set and test set differ. To solve such *domain adaptation* issues, the authors adopt the linearized GNN models and Central Moment Discrepancy (CMD) between (biased) training and i.i.d samples for regularization.

**Limitations And Societal Impact:**

Please consider addressing suggestions in the main review. In addition, I am willing to raise my score if the authors address my concern in the weakness section of the main review.

**Main Review:**

Strengths
====
The problem of addressing distributional shifts between training and test is important in machine learning but it is less addressed in the geometric (graph) machine learning. This paper has clear merit that tackles such problems with the simple approach SR-GNN and well-designed experiments. Experimental results show that the proposed SR-GNN clearly handles the problem of distribution shift in semi-supervised node classification tasks with imbalanced data.

Weakness
====
The only concern I have is about the fairness of evaluation. SR-GNN seems to use additional input $Z_{IID}$, which are the representations of unlabeled data sampled from the unifying set of training and test dataset. However, other baselines only consider input from the biased distribution. Is it a fair evaluation? I think authors should include additional baselines [1] that also consider features from unbiased samples to highlight the advantage of SR-GNN in the domain adaptation setting.

Suggestions
====
1. typo - line 62 (that that -> that)
2. It will be better to include axis labels in Figure 1. In addition, it seems ambiguous to see relations between F1 and CMD only with Figure 1. Would you consider adding any metric such as the Pearson correlation coefficient?

[1] Ganin et al., "Domain-Adversarial Training of Neural Networks", JMLR 2016.

**Time Spent Reviewing:**

10

---

> ### Author Response · Authors · 2021-08-10
> **Response to Reviewer 6hbj**
>
> Thank you for your comments and time spent reviewing our paper. We are happy that the reviewer appreciated our contributions in exposing the problems that the ML community is widely aware of in standard ML, but that have not been yet surfaced in the GNN world.
>
>
> ### 1.Fairness of the evaluation, SR-GNN used additional unlabeled IID Z.
> DANN is another way of achieving domain invariant representation learning, in addition to MMD/CMD/and other regularizations. Interestingly, a recent CMD paper has shown superior performance over DANN (https://arxiv.org/pdf/1702.08811.pdf please see Table 2).
> Nevertheless, the comparison you suggest is interesting -- please see in the Table below, comparing DANN and CMD on both GCN and APPNP, which we will add to the paper:
>
> |                      | Cora  | Cora    |     Citeseer  |               Citeseer     |   PubMed|                PubMed   |
>  | ------------ | ------------------------- | ------------------ | -------------- | ------------ | ----------------- | ------------ |
> |Method |                  micro-F1  | macro-F1     |    micro-F1 |  macro-F1  |        micro-F1 |  macro-F1
>  |GCN  |68.3  |67.2 | 62.4 | 60.2 | 59.2 | 53.8 |
>  |DANN | 69.8 | 68.5 | 63.8 | 61.0 | 64.8  |61.8 |
>  |CMD (Ours)  |71.0 | 69.4 | 65.0 | 62.3 | 67.5 | 66.2 |
>  |APPNP  |71.3  |69.2 | 63.9  |61.6 | 64.8 | 60.4 |
>  |DANN  |71.6  |69.5 | 64.3 | 61.8 | 67.8 | 65.4 |
>  |CMD (Ours)  |72.4  |70.1  |65.0 | 62.4  |70.4 | 68.7 |
>
> The hyper parameter λ is used to
> weight both the domain loss in DANN and our CMD regularization (Eq. 10). We tune this hyper parameter on
> the validation set and report models’ performance under the best λ (1 for Cora and Citeseer, 0.1 for
> PubMed). Two methods use the same amount of additional unlabeled data $Z_{IID}$. Notice that CMD (Ours) is the ablated method called (SR-GNN w.o. KMM) in the main paper – we note that we have already shown its performance can be further boosted using KMM.
>
> ### 2. Improving the visualization of Figure 1.
>
> For the clarity of Figure 1, we will use a fitted line to show the correlations and compute the Pearson coefficients on it.

---

> > ### Comment · Reviewer_6hbj · 2021-08-23
> > **Thanks for the response.**
> >
> > Thanks to the authors for the response and I really appreciate your sincere response.
> >
> > I am fully satisfied with your additional experiments against DANN, beyond the conceptual discussion. Now I believe that the suggested method has the clear merit that has not come from the additional unlabeled IID data. Thanks again for your effort!

---

> > > ### Author Response · Authors · 2021-08-25
> > > **Thanks.**
> > >
> > > Dear Reviewer 6hbj,
> > >
> > > Thanks again for your suggestions. We will definitely include the discussion and comparison between ours and DANN in the revised manuscript as well as the improved version of Figure 1.

---

### Official Review · Reviewer_HELz · 2021-07-15

**Rating:** 5
**Confidence:** 4

**Summary:**

Distributional shift is a common issue in machine learning.
In this paper, the authors study this topic in graph representation learning and also found the similar phenomenon.
They propose a framework called SR-GNN that is able to reduce influence of biased training data.
The authors have also conducted evaluation based on some benchmark datasets to show the effectiveness of the proposed method.

**Limitations And Societal Impact:**

Please refer to the above questions.

**Main Review:**

In general, the paper is written well and easy to follow. However, there are still several concerns that the authors need to address:

1. The practicality of the topic.
(1) Is there really exist distributional shift in the real world graph dataset, such as Cora, Citeseer, PubMed, ogb-arxiv datasets?
If the answer is yes, please use the some metrics, such as your paper shows in line 134, to compute the differences on these four datasets.
(2) All of the experiments in this paper were conducted on post-intervention datasets, which may not occur in real life.
In other words, I want to know whether using your method on the original Cora, Citeseer, PubMed and ogb-arxiv datasets can still improve performance.

2. The relationship between the label shift and the distributional shift.
Uneven labeling refers that there is no random or uniform labeling the training node on the graph.
Distributional shift means that there exits gap between training and test dataset distribution.
But is there any relationship between the above two?
Why does the deviation of the label lead to the deviation of the training set?
Please give detailed theoretical or experimental proofs.

3. Questions about the distributional shift.
(1) As shown in Figure1, I really want to know how to quantitatively change the difference between the distribution of the training set and the test set.
(2) What is the specific value of the distribution difference in the datasets in Table 1 and Table 2? Please give specific numerical indicators, such as CMD and so on.
(3) As shown in Figure3, does increasing alpha mean increasing the distribution gap?
If your answer is yes, please give some theoretical or experimental proofs.
If not, please add experiments similar to Figure 1, that is, when the distribution difference changes from small to large, your proposed method can alleviate the negative correlation between performance and distribution difference to a certain extent.
(4) Please refer to "GCN(IID)" in Table 2,3,4. I want to know how IID is guaranteed?

4. Other questions.
(1) In line 186: "Unfortunately, as there are no learnable layers at this stage in these models, one can not simply apply the distributional regularizer proposed in the previous section."
Why we can not use the final representation like standard GNN models to make regularization as equation (5) ?

5. A few minor errors.
(1) Table below the box line is missing, such as Table 2, 3.
(2) The tensor or matrix in the equation should be bolded.

**Time Spent Reviewing:**

5h

---

> ### Author Response · Authors · 2021-08-10
> **Response to Reviewer HELz**
>
> We thank the reviewer for thoughtful comments and time spent reviewing our paper.
>
> ### 1. On the practicality of robust GNNs
> You are bringing up an interesting point -- in academic node classification benchmarks, all nodes are labeled and thus acquiring IID training data is trivial!  This is also often the case for non GNN datasets (such as the standard ML UCI toy datasets).   However, we have **many** examples applying GNNs to **actual** real world datasets from industry where we have encountered non-IID training data.
>
> #### 1.1 Distributional shift in real world graph dataset.
>
> Please note that most academic datasets for GNNs (even “real world” ones like Cora, Citeseer, Pubmed, etc) are not representative of the datasets actually used in industry.  Some examples of (non-public) datasets which would contain non-IID labels would come from financial problems like fraud detection [1] or credit score prediction [2].
>
> [1] A Semi-supervised Graph Attentive Network for Financial Fraud Detection, ICDM 2019
>
> [2] Temporal-Aware Graph Neural Network for Credit Risk Prediction, SDM 2021
>
>
>
> #### 1.2 SR-GNN on original datasets like Cora, Citeseer.
>
> Because it is trivial to create IID training data on graphs which are fully labelled (like Cora, etc),  we would not expect to see much distribution shift on these datasets -- so there is nothing to fix.  However, if there are academic graphs which are labeled in a biased way, then we would expect our method to improve things.  We also expect our method to work well if the test data comes from a different distribution than the training data in general.  For example, in the molecular domain, one can imagine training on one class of molecule (organic compounds) and testing on another (plastics).
>
>
>
> ### 2. How does biased labeling create a difference between the training and testing feature distributions?
>
> Uneven labelling creates training data that is of different distribution than inference (IID) data. One way to think about it is: imagine we have a feature $f_i$ with values A, B, C, but we managed to label only the nodes with $f_i==A$. It is clear that our feature distributions for the training data != inference data.
>
> We note that uneven labelling is just one way of creating a distributional shift. Other sources of shift include simply using two different but related datasets for training and test (e.g in a standard Transfer Learning, you can be training on cats and dogs images but applying the model to horses and zebras).  Our method should also work in scenarios like this, but they are harder to simulate with open-source GNN datasets.
>
>
> ### 3. Questions about the distributional shift.
> The question of how to generate controlled biased samples for graphs is a broad and interesting one.  Here we have provided some of the first work in this area in order to study robustness in GNNs (our main contribution).  However a full analysis of this area is a much larger scope than one paper and is an interesting research on its own.  Nevertheless, we do our best to answer your questions here:
>
> #### 3.1  How to quantitatively change the difference between the distribution of the training set and the test set ?
> To change the distance between training and inference data, in Figure (1) we increase the training data labelling bias (so increasing choosing the nodes for training that are clustered vs IID spread throughout the graph). We use Personalized PageRank vectors (please see section 5.1) and set a minimum size of the PPR-neighbors to find such "clustered" nodes. Figure 1 demonstrates such biased samples from 3 datasets.
>
> #### 3.2 What is the specific value of the distribution difference in the datasets in Table 1 and Table 2?
> Please note that since the model's error on the unbiased data is a function of the error on the biased data and a function of the distance between biased and unbiased sample, one way to estimate this distance is to look at the GAP in model performance between IID model (GCN-IID) and GCN model (5th row in the table). We will add to this table a row that describes (in percent of IID vs biased sample) performance drop, so it is easier to judge which dataset is more biased.
>
>
> | |Cora-Micro-F1| Citeseer-Micro-F1 | PubMed-Micro-F1
> |-----|----|----|----|
> GCN (IID)| 80.8|70.3|79.8
> GCN (biased)|67.6|62.7|60.6
> Distance (%) |16.34% | 10.81% |24.06%
>
> Another way to view this is using the relative difference in CMD distance for each model.
>
>
> | |Cora CMD| Citeseer CMD | PubMed CMD
> |-----|----|----|----|
> GCN (IID)| 0.097|0.048|0.081
> GCN (biased)|0.124|0.050|0.298
> Difference (%) |28.07% | 5.39% |268.17%
>
>
> #### 3.3 Does increasing alpha mean increasing the distribution gap?
> In general, yes, $\alpha$ in personalized page-rank controls localization.  However, the clustering in the graph structure itself also affects the final bias of the generated sample.  This is present in Figure 3 where we illustrate that changing alpha changes performance (by controlling sample bias) for different methods, but doesn’t monotonically change the gap between SR-GNN and a non-robust baseline.
> We will prepare an additional analysis (similar to Figure 1) that shows how changing alpha changes CMD, but we will need additional time to run these experiments. (Figure 1 requires running hundreds of experiments.)
>
> #### 3.4 How IID is guaranteed in experiments?
> For all “IID” results: We perform 100 separate model trainings each using a different training label set. In each run, the nodes labeled for training are selected via uniform random sampling.
>
> ### 4. Why can we not use the final representation like standard GNN models to make regularization as equation (5) ?
> We intended for this comment to discuss where in a model the regularization should be applied.  If a model has no learnable layers $\Theta$ after the regularization, it will not affect any of them.  (For instance, if the regularization is applied directly to the features, there is nothing that can be learned.)   We highlight this distinction for linear GNN models like SGC[1]. We will refine this statement in the paper to clarify our intent.
>
> Thank you for pointing out formatting omissions/minor errors. We will fix them in the final version
>
> [1] Wu, Felix, et al. "Simplifying graph convolutional networks." International conference on machine learning. PMLR, 2019.

---

> > ### Comment · Reviewer_HELz · 2021-09-10
> > **Thanks for the response.**
> >
> > Dear authors,
> > Sorry for my late reply and thanks for the authors' extensive experiments.
> > My concerns are addressed well.
> > Best

---

### Official Review · Reviewer_i1qQ · 2021-07-16

**Rating:** 6
**Confidence:** 3

**Summary:**

Graph Neural Networks (GNNs) have been attracted much attention very recently because of surprising results in many applications. In general, GNNs assume that the training node labels follow the i.i.d. property. However, in practice, the labels could be very biased. To deal with the biased training nodes, the authors propose regularization methods that consider feature distance between training nodes and i.i.d sampled unlabeled nodes. CMD and KMM are key components of the regularization: CMD measures the distance between train nodes and general nodes, and KMM computes balanced weights of training loss functions. SR-GNN is the main framework that includes both CMD and KMM ideas. The experimental section shows that the biased data can hurt the performance, and SR-GNN can mitigate performance degradation.

**Limitations And Societal Impact:**

The authors discussed their limitations. Please refer to the main review comments for more discussions.

**Main Review:**

Originality:

This paper first considers the distributional shift for standard GNN models where training nodes are very biased. Since the problem is new, the proposed algorithm also has novelty.

Quality:

This paper is the first work that considers biased training nodes in GNN training. The proposed SR-GNN framework is intuitive, and the performance of SR-GNN is evaluated with three real graph data.

However, the experimental setting has some limitations.

First of all, the biased setting is artificial. To motivate this paper's bias problem, the authors should provide some real examples where the labeled nodes really have the biased property.

The main component of the proposed framework is not fully justified. Eq. (10) is the objective function of the proposed framework. In the objective function, weight $\beta_i$ and distance $d$ are crucial for the performance. Although there are many possible candidates for computing $\beta_i$ and $d$, this paper did not compare with other methods for finding $\beta_i$ and $d$.

As the authors also mentioned, it would be much nice to consider some more particular kinds of distribution shifts.

Clarity:

This paper is easy to read.

Significance:

This paper is the first work considering training bias in GNNs. I think many follow-up works are likely to refer to this work.


**Time Spent Reviewing:**

16

---

> ### Author Response · Authors · 2021-08-10
> **Response to Reviewer i1qQ**
>
> Thank you so much for your review.  We are very happy that you appreciated how we are the first to bridge two well developed topics - Transfer Learning and GNN - to address the real world problem of biased GNN training and highlighted the significance of our work.
>
> ### 1. Is this setting practically relevant?
> In most of the research papers on GNNs, people assume a perfect world with every node labeled. This is akin to the toy UCI machine learning datasets, where training and testing data are all sampled from the same distributions.  However, as we stressed in the introduction, a lot of real-world applications of GNNs (like abuse detection) do not enjoy this luxury.
>
> In other words, if a graph is fully labeled already, IID training data is trivial.  However, in most real world applications we actually do not have a completely labeled graph!  (Or anything close to one -- If we did have labels for every node in the graph, why would we bother trying to predict them?)  There can be additional complications with non-IID data in applications with temporal dependencies, like dynamic networks.
>
> In our improved draft, we will add a temporal dataset such as Reddit where data distribution could change over time.  We will also provide much more motivation in terms of real industry applications that we have observed this bias in.  To give a quick preview, biased data can arise when manual selection of nodes for labelling is involved; from delayed labelling (e.g. conversion attribution); automatic labelling that uses some pre-existing (but not all-encompassing heuristics), weakly connected graphs, class unbalanced graphs, and more.
>
> ### 2. Questions about methods for choosing $\beta$ and $d$.
>
> First, we’d like to clarify that the $\beta_i$ are the weights we find from the training data, and as such, they are not hyperparameters, per se.  To obtain the values of $\beta_i$, we specify the form of the quadratic problem in Eq.9, which is convex. Hence, any quadratic solver can get the exact solution for $\beta$. We use cvxopt in our experiment.  For the lower and upper bound of $\beta$ (the constraints added in (9)), we provide the sensitivity study in Figure 4.b (which basically demonstrates that the results are not too sensitive). You are correct that other instance weighting optimizations could be proposed, but we leave that to explore for future work.
>
> Second, for the distance metric used in distributional regularizer, indeed, various distances measures can be adopted (e.g. MMD, KL or even domain adversarial heads[1]). We chose CMD because it has been shown to outperform other metrics and be more robust to the weight with which it is added ($\lambda$) (see: https://arxiv.org/pdf/1702.08811.pdf).
> We also provide the additional result of adopting domain adversarial heads (DANN[1]) as d on GCN and APPNP. Although adversarial heads also improve the model robustness under distributional shifts, CMD yields better performance.
>
> |                      | Cora  | Cora    |     Citeseer  |               Citeseer     |   PubMed|                PubMed   |
>  | ------------ | ------------------------- | ------------------ | -------------- | ------------ | ----------------- | ------------ |
> |Method |                  micro-F1  | macro-F1     |    micro-F1 |  macro-F1  |        micro-F1 |  macro-F1
>  |GCN  |68.3  |67.2 | 62.4 | 60.2 | 59.2 | 53.8 |
>  |d=DANN | 69.8 | 68.5 | 63.8 | 61.0 | 64.8  |61.8 |
>  |d=CMD |**71.0** | **69.4** | **65.0** | **62.3** | **67.5** | **66.2** |
>  |APPNP  |71.3  |69.2 | 63.9  |61.6 | 64.8 | 60.4 |
>  |d=DANN  |71.6  |69.5 | 64.3 | 61.8 | 67.8 | 65.4 |
>  |d=CMD |**72.4**  |**70.1**  |**65.0** | **62.4**  |**70.4** | **68.7** |
>
>
> We vary the number of moments k used in CMD to better understand its behavior (Figure 4.a.). When the K=1, the distance d is simply the mean distance between training and unlabeled IID representation. The number of moments to use will depend ultimately on how different feature distributions are between the training and unbiased/inference sample. For a more pronounced bias, higher order moments will be needed to match the distributions.
>
>
> [1] Ganin et al., "Domain-Adversarial Training of Neural Networks", JMLR 2016.

---

> > ### Comment · Reviewer_i1qQ · 2021-08-22
> > **Re**
> >
> > Thank you for the reply. I hope the manuscript can be revised by reflecting all the concerns and answers.

---

### Official Review · Reviewer_RfY9 · 2021-07-16

**Rating:** 7
**Confidence:** 4

**Summary:**

The paper studies the influence of biases in training data that are due to distributional shifts on the use of GNNs for semi-supervised learning, and proposes a new method to tackle this bias issue, i.e., the Shift-robust GNN (SR-GNN). SR-GNN is shown to perform better than standard GNNs in the presence of distributional-related bias.

**Limitations And Societal Impact:**

The paper addresses certain general limitations of the method, but does not discuss societal impact. How unbalanced are the classes in the examples used? Could the authors add a last column in A.1 to show the percentage of labels per class?

**Main Review:**

GNNs, as many ML approaches, suffer from overfitting to the distribution of the training data. This typically arises in graph-based approaches when the nodes used for labeling in SSL are not IID.

As the goal is to use GNNs for SSL, a discussion on the connectedness of the graph would be useful. In the example from the paper on spam detection, depending on the dataset size, the graph might not be fully connected if the two classes are very far apart in the representation space $Z$. Would that be a problem? If yes, how to avoid it?

There are many different types of distributional shift, but the one considered here seems to be tightly related to the covariate shift where only the distribution of the covariates (here one can think of the representation $Z$ instead of the input $X$) varies while the conditional stays the same. I don’t think there is any reference in the paper on this. Could the authors extend on this similarity, or if not, the differences? Would the authors have a reference for the representation shift as defined on L132-133? Could this approach be used for other types of distributional shifts?

The locality aspect is not very clear to me. Would this be equivalent to sampling in the presence of unbalanced classes? What is the difference between the two? The locality aspect makes me think in spatial terms rather than imbalance.

Other comments:

•	L21: varying exactly – not straightforward, maybe “decide”

•	L23: difference

•	L26: most works

•	L27: which “this”?

•	Both distribution shift and distributional shift are used

•	L66: propose a

•	L87: why unlabeled in brackets?

•	L98: to

•	L102: F not defined

•	$M$ is used both for the training data size and the labeled data size. In SSL, I understand the training dataset to be larger than the labeled dataset so the two should be different. Maybe I am missing smth?

•	L140: $K$ -> $k$

•	L148: $z_M$

•	Fig. 1: ideally add labels for the axis. For the Cora dataset it’s difficult to see that the performance degrades as a function of CMD, it looks fairly uniformly distributed to me.

•	CMD: why not combine all moments under the sum, why keep the first moment separate, it would be more intuitive to use $c_1$ instead of expectation. How easy/difficult is it to compute CMD?

•	L170: what do the ‘ represent?

•	L179: distinct different

•	L183: should it be $A^K$ instead of k?

•	Eq 8 and 10: is there a $\sum$ missing?

•	Eq 9 : missing $\beta_i$ in the second term? Should the min be over all $\beta$, not just $\beta_i$?

•	L198: sum is over i not $\alpha_i$?

•	L203: what is $l_i$? should it be $y_i$?

•	L223: repeatable -> reproducible?

•	L234-238: How is the number of nodes needed per label known in advance (=enough number of nodes per label)?

•	Could you describe a bit the APPNP model and ideally define all the acronyms before using (DGI, GAT, etc)?

•	L271: how is $h_i$ used here?

•	Define Micro-F1 and Macro-F1

•	Check spelling: lineared, ppr -> PPR

•	Notations: check for A, E, F, \Theta – sometimes bold, sometimes normal


Originality: the paper addresses an important aspect of generalization/extrapolation and suggests a solution to address the bias in SSL using GNNs. The work looks fairly original to me even though I am not an expert in the field.

Quality: the paper is of good quality with a clear explanation of the theory and clean and thorough experiments.

Clarity: the paper is mostly clearly written with a few exceptions as in my comments above

Significance: Biases and extrapolation are important challenges and solutions to address them as in the current paper are relevant


**Time Spent Reviewing:**

4h

---

> ### Author Response · Authors · 2021-08-10
> **Response to Reviewer RfY9**
>
> Thank you so much for a very detailed review and all the comments about notations and suggested fixes to improve the clarity & readability!  We will directly address them all in the revised draft. We are very happy that you appreciated how we were the first to bridge two well developed topics - Transfer Learning and GNN - to address the real world problem of biased GNN training.
>
> ### 1. How is biased training data related to the connectedness of the graph?
> The reviewer brings up a great point that the connectedness of the graph is closely related to the distributional shift: variance in feature distributions in different components is indeed one source of distributional shift. If only some graph components are sampled in a graph with homophily, we will end up with biased training data. In a spam detection graph, positive and negative labels can be present in different connected components. By introducing randomly sampled IID nodes, our hope is that such a sample will include nodes from all major components. To restate -- a biased sample of training nodes may ignore major parts of a graph, while an IID sample helps the model generalize to the average conditions found in the graph.
>
>
> ### 2. What’s the relation between distributional shift and covariate shift?
> You are exactly right - the assumptions for our distributional shift are the same as the covariate shift assumptions (i.e. feature distributions are different between the training and unbiased sample, and the conditional p(y|x) distribution is the same). If the conditional distributions are different, we still can apply our method (to match the feature distributions) and then, as a final step, fine tune the model’s classification heads to account for different y|x. This, however, will require some LABELED unbiased sample, which in our setting we didn't assume we had (we used unlabeled unbiased sample for our intervention). For more details on representation shift (including definitions) please see https://arxiv.org/pdf/1812.11806.pdf, Section 4.2.
> One additional complication in the GNN setting is that bias accumulates between the layers, since the graph structure is (directly) applied in each layer!  This can result in amplification of the initial feature distribution shift the deeper in the network we go.
>
>
>
> ### 3. The locality aspect is not very clear to me. The locality aspect makes me think in spatial terms rather than imbalance.
>
> We are a little confused by the question, but will attempt to answer.  Many datasets have the property that spatial locality in a graph provides some “closeness” in the node’s feature distribution, a property often called homophily.  So we can mimic one kind of bias found in training data by expanding locally around a particular seed node and using these as the examples for one class’s labels.  A similar process frequently occurs when human raters are asked to label a graph.  They find a node with a given label and inspect its immediate neighbors (in a kind of breadth-first search) to create more labeled data. The distribution of different classes will not be changed in localized training data if we keep the class balanced in Sec 5.1.
>
> We’re happy to follow up if this wasn't a satisfactory enough explanation.
>
> ### 4. Societal/Broader Impact?
>
> Indeed, you bring up a very good point, the methods we use for regularization (e.g. CMD, MMD) have been extensively used for aiding fairness, thus we envision that our bias reduction techniques can also be applied to increase the fairness of GNN models in particular domains. The only change this will entail is a redefinition of a domain: going from biased and unbiased domains to the domains of interest (based on various groups of data).
>
> ### 5.How unbalanced are the classes in the examples used?
>
> Many of the GNN datasets (Cora, etc) are quite balanced. The exception is ogb-arxiv, but we only used frequent labels in study to alleviate balance concerns.  We will add the exact class label distributions (histograms) for each dataset to the appendix.
>
> ### 6. Other minor comments about clarity.
> 6.1 The F in line 102 denotes the size of node features.
>
> 6.2 M is used for both training and labeled data size. In our paper, we intend to define training nodes as if they are labeled. In SSL on graphs, every node can be used during GNN training.
>
> 6.3 L234-238, we use 20 labels per class for three citation networks as a common practice. For ogb-arxiv, we use 1% and 5% label ratio to compute the number of training data for each class (i.e. keep the original class distribution).
>
> 6.4 We plan to update Figure 1 with a fitted line to show the correlations and compute the Pearson coefficients. For Cora, we will add more bias into the training for a more consistent result.
>
> 6.5 We will update the first-order momentum of the CMD calculation as $c_1$. For a given k, the computation is linear to the total number of samples used O(N(a+b)).
>
> 6.6 In line 183, $A^k$ denotes the k round of propagation in the graph. We will add the “no learnable layer” description to $F_1$ for clarity.
>
> 6.7 In Eq.8 and Eq.10, we miss the $\sum$. In line 198, it’s $\sum$ over i. In line 203, $l_i$ should be $y_i$. Thanks for pointing these out.
>
> 6.8 In line 271, we use $h_i$ to introduce the dimension of the hidden representation n.

---

### Decision · Program_Chairs · 2021-09-27

**Decision:**

Accept (Poster)

**Comment:**

This paper studies the influence of biases in training data that are due to distributional shifts on the use of GNNs for semi-supervised learning. The key idea is to adopt the linearized GNN models and Central Moment Discrepancy (CMD) between (biased) training and i.i.d samples for regularization. This paper proposes a new method to tackle this bias issue, i.e., the Shift-robust GNN (SR-GNN). SR-GNN is shown to perform better than standard GNNs in the presence of distributional-related bias.

However, there exists some limitations as follows.

1) The practicality: does there really exist distributional shift in the real world graph dataset, such as Cora, Citeseer, PubMed, ogb-arxiv datasets? Meanwhile, all of the experiments in this paper were conducted on post-intervention datasets, which may not occur in real life.

2) The relationship: Uneven labeling refers that there is no random or uniform labeling the training node on the graph. Distributional shift means that there exits gap between training and test dataset distribution. Is there any relationship between the label shift and the distributional shift? Why does the deviation of the label lead to the deviation of the training set?

3) The shift: how to quantitatively change the difference between the distribution of the training set and the test set. What is the specific value of the distribution difference in the datasets in Table 1 and Table 2? Does increasing alpha mean increasing the distribution gap? If your answer is yes, please give some theoretical or experimental proofs. If not, please add experiments similar to Figure 1, that is, when the distribution difference changes from small to large, your proposed method can alleviate the negative correlation between performance and distribution difference to a certain extent.

This paper is a boardline case according to the average rating. While the reviewers had some concerns on the significance, the authors did a particularly good job in their rebuttal. Thus, all of us have agreed to marginally accept this paper for publication! Please include the additional experimental results in the next version.